# A Human ABC Transporter *ABCC4* Gene SNP (rs11568658, 559 G > T, G187W) Reduces ABCC4-Dependent Drug Resistance

**DOI:** 10.3390/cells8010039

**Published:** 2019-01-10

**Authors:** Megumi Tsukamoto, Miho Yamashita, Tsuyoshi Nishi, Hiroshi Nakagawa

**Affiliations:** 1Department of Applied Biological Chemistry, Graduate School of Bioscience and Biotechnology, Chubu University, 1200 Matsumoto-cho, Kasugai 487-8501, Japan; gr16801@isc.chubu.ac.jp (M.T.); fr15109-5214@sti.chubu.ac.jp (M.Y.); 2Department of Biomolecular Science and Regulation, Osaka University, Ibaraki, Osaka 567-0047, Japan; tnishi@sanken.osaka-u.ac.jp

**Keywords:** ATP-binding cassette (ABC) transporter, ABCC4, MRP4, SNP

## Abstract

Broad-spectrum drug resistance is a major obstacle in cancer treatment, which is often caused by overexpression of ABC transporters the levels of which vary between individuals due to single-nucleotide polymorphisms (SNPs) in their genes. In the present study, we focused on the human ABC transporter ABCC4 and one major non-synonymous SNP variant of the *ABCC4* gene in the Japanese population (rs11568658, 559 G > T, G187W) whose allele frequency is 12.5%. Cells expressing ABCC4 (G187W) were established using the Flp-In™ system based on Flp recombinase-mediated transfection to quantitatively evaluate the impacts of this non-synonymous SNP on drug resistance profiles of the cells. Cells expressing ABCC4 (WT) or (G187W) showed comparable *ABCC4* mRNA levels. 3-(4,5-Dimethyl-2-thiazol-2-yl)-2,5-diphenyl-2*H*-tetrazolium bromide (MTT) assay indicated that the EC_50_ value of the anticancer drug, SN-38, against cells expressing ABCC4 (G187W) was 1.84-fold lower than that against cells expressing ABCC4 (WT). Both azathioprine and 6-mercaptopurine showed comparable EC_50_ values against cells expressing ABCC4 (G187W) and those expressing ABCC4 (WT). These results indicate that the substitution of Gly at position 187 of ABCC4 to Trp resulted in reduced SN-38 resistance.

## 1. Introduction

Broad-spectrum drug resistance, which is often due to ABC transporters, is a major obstacle in cancer treatment. The level of drug resistance of cancer cells is usually related to the expression levels of ABC transporters and is complicated by single-nucleotide polymorphisms (SNPs) in their genes. In fact, various quantitative analyses of ABC transporter function (WT or SNPs) showed that SNPs in ABC transporter genes modify their expression levels, transport function, and/or intracellular distribution [1,2,3,4,5,6,7,8,9,10]. Previous quantitative studies revealed the impacts of SNPs in ABC transporter genes on drug sensitivity of the cells [7,8,9,10,11].

The *ABCC4* gene, which is located on chromosome 13q32.1 and encodes the 1325-amino acid human ABCC4 (MRP4), is one of the 48 ABC transporters expressed in various tissues, including the liver, kidney, ovary and blood cells [3,12,13]. ABCC4 has been reported to transport antiviral (e.g., Azidothymidine [14], Ganciclovir [15] and Nelfinavir [16]), antibiotic (e.g., Cefazolin [17] and Ceftizoxime [17]), antihypertensive (e.g., Furosemide [18] and Hydrochlorothiazide [18]) and anticancer (e.g., azathioprine [10], 6-mercaptopurine [2] and SN-38 [19]) drugs. Some of the ≥140 non-synonymous SNPs in the *ABCC4* gene (rs45454092, 551 T > A, M184K; rs11568658, 559 G > T, G187W; rs200387797, 890 A > G, N297S; rs2274407, 912 G > T, K304N; rs11568705, 1208 C > T, P403L; rs753414892, 1167 A > G, Y556C; rs11568668, 1460 A > G, G487E; rs3765534, 2269 G > A, E757K; rs146708960, 2326 G > A, V776I; and rs11568644, 3425 C > T, T1142M) have been quantitatively shown to alter the affinity of ABCC4 for its substrate drugs [2,3,6,20] and/or the drug resistance profiles of the cells in which it is expressed [10]. According to the Human Genetic Variation Database and Integrative Japanese Genome Variation Database, rs11568658 (G187W), rs2274407 (K304N) and rs3765534 (E757K) are major *ABCC4* gene SNPs in the Japanese population (Table 1). As the impact of rs11568658 (G187W) on drug resistance profiles of cells remains to be elucidated, we established cells expressing ABCC4 (G187W) based on the Flp recombinase-mediated transfection system (Flp-In™ system) and quantitatively evaluated the impacts of rs11568658 (G187W) on drug resistance profiles of the cells.

## 2. Materials and Methods

### 2.1. Chemicals and Biological Reagents

The following reagents and drugs were purchased from the commercial sources indicated in parentheses: antibiotic-antimycotic cocktail solution, l-glutamine, high-glucose Dulbecco’s modified Eagle’s medium (DMEM), hygromycin B (Nacalai Tesque, Inc., Kyoto, Japan); fetal bovine serum (FBS) (Equitech-Bio, Inc., Kerrville, TX, USA); and 3-[4,5-dimethylthiazol-2-yl]-2,5-diphenyltetrazolium bromide (MTT reagent) (Sigma-Aldrich Co., St. Louis, MO, USA). SN-38 was generously provided by Yakult Honsha Co., Ltd. (Tokyo, Japan). All other chemicals used were of analytical grade.

### 2.2. Preparation of Plasmids Carrying the ABCC4 (G187W) Variant cDNA

The pcDNA5/FRT expression vector carrying *ABCC4* (*G187W*) cDNA (Figure 2A) was prepared from pcDNA5/FRT/ABCC4 (WT) vectors that we prepared previously [10]. The nucleic acid sequence of the *ABCC4* (*G187W*) gene was obtained from the NCBI dbSNP database. Briefly, non-synonymous SNP variants of ABCC4 were generated by site-directed mutagenesis using the PrimeSTAR^®^ Max DNA Polymerase (Takara Bio Inc., Otsu, Japan) with the following primers: forward, 5′-cttagtaacatggccatgtggaagacaaccacaggc-3′; reverse, 5′-gcctgtggttgtcttccacatggccatgttactaag-3′. After polymerase chain reaction (PCR) (10 s at 98 °C, 12 cycles of 10 s at 98 °C, 15 s at 55 °C and 10 min at 72 °C), the reaction mixture was treated with DpnI endonuclease for 1 h at 37 °C to degrade the original template plasmid pcDNA5/FRT/ABCC4 (WT) vectors. The nucleic acid sequence of *ABCC4* (*G187W*) cDNA in the resulting amplicons was confirmed by sequencing with Applied Biosystems 3130 and 3130xl genetic analyzers (Applied Biosystems, Foster City, CA, USA) as shown in Figure 2B.

### 2.3. Cell Culture

Flp-In-293 cells (Invitrogen) were maintained in high-glucose DMEM supplemented with 10% (*v*/*v*) heat-inactivated FBS (Equitech-Bio), 4 mM l-glutamine, 100 U/mL penicillin, 100 μg/mL streptomycin, 250 ng/mL amphotericin B and 100 mg/mL zeocin in a humidified atmosphere of 5% (*v*/*v*) CO_2_ in air. The number of viable cells was determined with the Trypan Blue dye exclusion method using a hemocytometer. The Flp-In-293 cells were transfected with the pcDNA5/FRT/ABCC4 (G187W) expression vectors and the Flp recombinase expression plasmid pOG44 by using LipofectAmine™-2000 (Invitrogen) according to the manufacturer’s instructions. Colonies resistant to 50 mg/mL hygromycin B solution (Nacalai Tesque) were picked and sub-cultured. The resulting cells incorporating pcDNA5/FRT/ABCC4 (G187W) were designated as Flp-In-293/ABCC4 (G187W) cells and were maintained in high-glucose DMEM supplemented with 10% (*v*/*v*) heat-inactivated FBS, 4 mM l-glutamine, 100 U/mL penicillin, 100 μg/mL streptomycin, 250 ng/mL amphotericin B and 50 mg/mL hygromycin B in a humidified atmosphere of 5% (*v*/*v*) CO_2_ in air.

### 2.4. Preparation of Total RNA and Synthesis of First-Strand cDNA

Cells were seeded into 35-mm dishes (TrueLine) at a density of 1 × 10^6^ cells/well and pre-cultured for 3 days. The cells were then collected by pipetting with culture medium into 1.5-mL tubes followed by centrifugation at 300× *g* for 5 min at 4 °C. The resulting cell pellets were rinsed twice with 1 mL of PBS(−) and suspended in 600 μL of lysis buffer prepared from Lysis/binding buffer (Roche Diagnostics, Mannheim, Germany) and stored at −80 °C until use for preparation of total RNA.

Total RNA was extracted from the cell lysates using a High Pure RNA Isolation Kit (Roche Diagnostics) according to the manufacturer’s instructions. The concentration and quality of total RNA in each extract were measured spectrophotometrically and evaluated by calculating the A260/A280 values, respectively (DU640; Beckman Coulter, Fullerton, CA, USA). Thereafter, the total RNA was reverse transcribed using a High-Capacity cDNA Reverse-Transcription Kit (Thermo Fisher Scientific Inc., Waltham, MA, USA) and random hexamers as a primer according to the manufacturer’s instructions.

### 2.5. Quantitative Real-Time PCR

*ABCC4* mRNA levels in the cells were measured quantitatively using GoTaq^®^ qPCR Master Mix, 2× (Promega, Tokyo, Japan) and gene-specific primers, for ABCC4 (forward primer, 5095-085 (Takara Bio), reverse primer, 5095-086 (Takara Bio)) and the housekeeping gene GAPDH as an internal control (forward primer, 10000459 (Takara Bio), reverse primer, 20000459 (Takara Bio)) with an Applied Biosystems 7500 Fast Real-Time PCR System (Thermo Fisher Scientific Inc., Waltham, MA, USA).

### 2.6. MTT Assay

MTT assay was performed as previously described [10]. In facts, cells were seeded into 96-well plates (Thermo Fisher Scientific) at a density of 5 × 10^5^ cells/well and pre-cultured for 24 h. The cells were then exposed to different concentrations of the anticancer drugs (azathioprine, 6-mercaptopurine and SN-38) for 72 h according to the results reported previously [10], where the highest final concentrations of these drugs were 100 nM (SN-38) or 100 μM (azathioprine and 6-mercaptopurine) and the lowest was 0 M (control). After 72 h of treatment with each drug, the cells were incubated in the presence of 500 μg/mL MTT for a further 3 h. Thereafter, 100 μL of 20% sodium dodecyl sulfate (SDS) was added to each well and the plates were incubated overnight at 37 °C in a humidified atmosphere of 5% (*v*/*v*) CO_2_ in air. The absorbance of formazan, a metabolite of MTT, in each well of the resulting solution was measured photometrically at 570 nm and at a reference wavelength of 630 nm using a Thermo Lab systems Multiskan Jax (Thermo Fisher Scientific Inc., Waltham, MA, USA). Cell viability was expressed as a percentage relative to control. Cytotoxicity was assessed with reference to the EC_50_ value, which was defined as the concentration required for 50% reduction of viability based on the survival curve.

### 2.7. Preparation of Cell Lysates for SDS-PAGE

Cell lysates for SDS-PAGE was prepared as previously described [10]. In facts, cells were seeded into 35-mm dishes (TrueLine, Baton Rouge, LA, USA) at a density of 1 × 10^6^ cells/well and pre-cultured for 3 days. After pre-culture, the cells were collected by pipetting with culture medium into 1.5-mL tubes and centrifuged at 300× *g* for 5 min at 4 °C. The resulting cell pellets were rinsed twice with 1 mL of PBS(−) and suspended in lysis buffer (50 mM Tris-HCl, pH 7.6, 5 mM EDTA, pH 8.0, 120 mM NaCl, 1% Triton X-100, 1 mM DTT, protease inhibitor and phosphatase inhibitor). Thereafter, the cell lysates were centrifuged at 800× *g* for 10 min at 4 °C. The resulting supernatants were retained and the protein contents of the samples were determined by the Bradford method. After this assay, 50 μg of the supernatant protein was treated with PNGase F for 10 min at 37 °C.

### 2.8. SDS-PAGE and Western Blotting

SDS-PAGE was performed as previously described [10]. In facts, cell lysate samples were prepared independently from each group of cells in triplicate and aliquots of 5 μg were mixed. The mixtures were fractionated by SDS-polyacrylamide gel electrophoresis (PAGE) (7.5%) and the gels were transferred onto nitrocellulose membranes (GE Healthcare UK Ltd., Bucks, UK). The membranes were soaked in blocking buffer consisting of TBST (50 mM Tris-HCl, 150 mM NaCl and 0.05% (*v*/*v*) Tween 20) containing 5% (*w*/*v*) skim milk powder at room temperature for >1 h and left to stand overnight at 4 °C. The membranes were incubated with monoclonal anti-ABCC4 antibody (M4I-10; GeneTex Inc., Alton Parkway Irvine, CA, USA) or anti-GAPDH antibody (anti-GAPDH-Clone 6C5 mouse monoclonal, igG2b; American Research Products, Inc., Waltham, MA, USA) at 1:1000 dilution in TBST containing 5% (*w*/*v*) skim milk powder for 1 h at room temperature with shaking after rinsing with TBST. Thereafter, the membranes were washed with TBST, followed by incubation with appropriate secondary antibody for 1 h at room temperature with shaking. The secondary antibody for ABCC4 was rabbit anti-Rat IgG:HRP (Enzo Life Sciences, Inc., Farmingdale, NY, USA) and that for GAPDH was HRP-conjugated anti-mouse IgG antibody (Cell Signaling Technology, Inc., Danvers, MA, USA) at 1:1000 dilution in TBST containing 5% (*w*/*v*) skim milk powder. The blots were developed using Western Lighting Chemiluminescent Reagent Plus (PerkinElmer Life and Analytical Sciences, Boston, MA, USA) and detected with WSE-6100 LuminoGraph I (Atto Corp., Tokyo, Japan). The signal intensities derived from ABCC4 or GAPDH were determined using ImageJ (Wayne Rasband, Bethesda, MD, USA).

### 2.9. Statistical Analysis

Statistical analyses were performed using JSTAT version 20.0J (Masato Sato, Japan), which is a software for statistical One-way ANOVA and Tukey’s honestly significant difference (HSD) test analysis. In all analyses, *p* < 0.01 was taken to indicate statistical significance.

## 3. Results

### 3.1. Levels of ABCC4 mRNA and Protein in Cells Expressing ABCC4 (G187W)

We used Flp-In-293 cells corresponding to the Flp-In™ system to establish cells expressing ABCC4 (G187W) (Figure 1; Figure 2). Flp-In-293 cells were transfected with the *ABCC4* (*G187W*) cDNA, which integrated into the FRT-tagged genomic DNA and then selected using hygromycin B. The resulting hygromycin B-resistant cells were analyzed by qPCR to determine the *ABCC4* and *GAPDH* mRNA levels and compared with Flp-In-293/Mock and ABCC4 (WT) cells to confirm the proper functioning of the Flp-In™ system. In the present study, total RNA was prepared from Flp-In-293/Mock, ABCC4 (WT) and ABCC4 (G187W) cells and their qualities were evaluated by comparing the A260/A280 values of total RNA and the threshold cycles in qPCR among samples. Since those were comparable (data not shown), the levels of *ABCC4* mRNA were subsequently normalized relative to those of *GAPDH* and compared. As shown in Figure 3, *ABCC4* mRNA levels in cells transfected with *ABCC4* cDNA were >23-fold higher than those in Flp-In-293/Mock cells, as we reported previously [10]. In contrast, the levels of *ABCC4* mRNA were comparable between Flp-In-293/ABCC4 (WT) and (G187W) cells, indicating that the Flp-In™ system was functioning in the cells established in the present study.

As qPCR clearly showed that the Flp-In™ system functioned in Flp-In-293/ABCC4 (G187W) cells, western blotting analysis was performed to evaluate ABCC4 and GAPDH expression in the cells after treatment with PNGase F to remove the glycomoieties on ABCC4. As shown in Figure 4, the level of ABCC4 expression corresponded to that of *ABCC4* mRNA and the levels of ABCC4 in Flp-In-293/ABCC4 (WT) and (G187W) cells were much higher than that in Flp-In-293/Mock cells. In contrast, the levels of ABCC4 in cells expressing ABCC4 (G187W) were comparable to those in cells expressing ABCC4 (WT).

### 3.2. Anticancer Drug Resistance of Cells Expressing ABCC4 (G187W)

MTT assay was performed to examine the anticancer drug resistance of Flp-In-293/ABCC4 (G187W) cells and the results were compared among Flp-In-293/Mock, ABCC4 (WT) and ABCC4 (G187W) cells. As described in Materials and Methods, Flp-In-293/ABCC4 (G187W) cells were examined for resistance to azathioprine, 6-mercaptopurine and SN-38 and the results are summarized as EC_50_ values in Table 2. Cells expressing ABCC4 (WT) showed greater resistance to azathioprine, 6-mercaptopurine and SN-38 than Flp-In-293/Mock cells (Table 2, Figure 5) as we reported previously [10]. According to the EC_50_ values, cells expressing ABCC4 (WT) showed 4.2-, 4.4- and 7.6-fold greater resistance to azathioprine, 6-mercaptopurine and SN-38 than Flp-In-293/Mock cells, respectively (Table 2). Flp-In-293/ABCC4 (G187W) cells also showed resistance to azathioprine, 6-mercaptopurine and SN-38 compared to Flp-In-293/Mock cells (Table 2 and Figure 5). According to the EC_50_ values, Flp-In-293/ABCC4 (WT) cells showed 1.84-fold greater resistance to SN-38 compared to Flp-In-293/ABCC4 (G187W) cells (Table 2).

## 4. Discussion

### 4.1. Establishment of Human ABCC4 (G187W)-Expressing Cells Using the Flp-In™ System

The drug resistance of cancer cells and the individual differences in their levels are usually due to the overexpression of ABC transporters and SNPs in their genes, respectively [8]. We found that the drug sensitivities of cells expressing ABCC4 were modified by the first (rs2274407 (K304N)) and second (rs3765534 (E757K)) most common SNPs in the *ABCC4* gene present in the Japanese population (Table 1) [10]. In the present study, we established cells expressing the third most common *ABCC4* gene SNP in Japanese population (rs11568658 (G187W)) (Table 1) based on the Flp-In™ system, which can be sued to integrate a single copy of a cDNA into the FRT site prepared in the telomeric region of the short arm on one copy of chromosome 12 in Flp-In-293 cells [7] and quantitatively evaluated the impacts of this SNP on drug resistance of the cells. Although the integration site and the *ABCC4* cDNA copy number were not determined in the present study, the levels of *ABCC4* mRNA expression were comparable between Flp-In-293/ABCC4 (WT) and (G187W) cells (Figure 3). These results suggested that the transfected *ABCC4* cDNA was integrated into the designated site in the chromosome of the cells and that the integrated *ABCC4* cDNA copy number was the same in Flp-In-293/ABCC4 (WT) and (G187W) cells.

### 4.2. Anticancer Drug Resistance of Flp-In-293/ABCC4 (G187W) Cells

ABCC4 has been reported to transport a broad spectrum of xenobiotics, including antiviral, antibiotic, antihypertensive and anticancer drugs [2,3,6,13,14,15,19,21,22,23,24,25,26,27]. Consistent with previous studies from our and other laboratories [2,6,10,19,24,25,27], cells expressing ABCC4 (WT) exhibited drug resistance to substrate anticancer drugs for ABCC4 (azathioprine, 6-mercaptopurine and SN-38) compared to Flp-In-293/Mock cells as we previously reported [10]. The *ABCC4* (*G187W*) variant cDNA was successfully prepared as shown in Figure 2 and cells expressing ABCC4 (G187W) exhibited drug resistance to substrate anticancer drugs for ABCC4 (azathioprine, 6-mercaptopurine and SN-38) compared to Flp-In-293/Mock cells. These observations indicated that *ABCC4* (*G187W*) cDNA transfected into Flp-In-293 cells was functional and we could examine the effects of the SNP (rs11568658, 559 G > T, G187W) in the *ABCC4* gene on ABCC4-mediated drug resistance in transfected cells, although ABCC4 is intrinsically expressed in Flp-In-293 cells [10,28] and the transfected *ABCC4* (*G187W*) cDNA may alter the expression levels of other ABC transporters.

The drug resistance profiles of cells expressing ABCC4 (G187W) have not been examined previously, although Banerjee et al. (2016) established cells expressing ABCC4 (G187W). They prepared cells expressing ABCC4 (G187W) using a traditional transfection system with pcDNA3.1 vector, where the copy number and integration site of the cDNA into the chromosome were not controlled and the resulting cells were not suitable to examine the effects of *ABCC4* gene SNPs on ABCC4-mediated drug resistance in cells [20]. On the other hand, we established cells expressing the wild-type (WT) or SNP variants of human ABCC4 with controlled copy number and site of cDNA integration into the chromosome and showed that the drug resistance profiles of cells expressing the SNP variants of ABCC4 were significantly different from those of cells expressing ABCC4 (WT) [10]. The drug resistance profiles of cells were reported to be affected by the levels of ABC transporter expression [29,30]. Our previous studies showed that non-synonymous SNPs in the *ABCG2* gene can affect drug resistance of cells by altering their substrate specificity, intracellular localization, or intracellular stability under conditions where the mRNA levels are comparable [4,7,9]. Banerjee et al. reported that ABCC4 (G187W) altered the affinity of ABCC4 for MMA(GS)2 in a protein-based assay but did not affect intracellular localization [20]. In the present study, the expression levels of ABCC4 (G187W) in Flp-In-293/ABCC4 (G187W) cells were comparable to those in Flp-In-293/ABCC4 (WT) cells. Thus, ABCC4 (G187W) would increase intracellular accumulation levels of test drugs by altering the substrate specificity of ABCC4. It is important to measure the levels of intracellular accumulation of test drugs to understand the molecular mechanisms by which substitution of Gly187 to Trp in ABCC4 alters the drug resistance profiles of cancer cells. Such studies are currently in progress in our laboratory.

In summary, we quantitatively evaluated the impact of rs11568658 (G187W) on the drug resistance profiles of cells using our previously proposed easy-to-use and quantitative approach [10]. The results presented here will improve our understanding of the effects of non-synonymous SNPs in the *ABCC4* gene on cancer chemotherapy and could contribute to the development of novel therapeutic strategies as well as diagnostic and therapeutic approaches for cancer chemotherapy.

## Figures and Tables

**Figure 1 cells-08-00039-f001:**
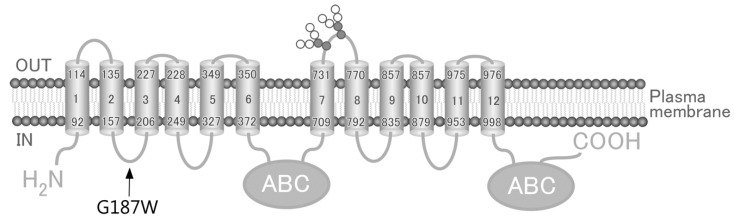
Schematic illustration of human ABCC4 and the location of SNP (rs11568658, 559 G > T, G187W). Arrow, location of SNP (rs11568658, 559 G > T, G187W); ABC, ATP binding cassette (nucleotide binding domain).

**Figure 2 cells-08-00039-f002:**
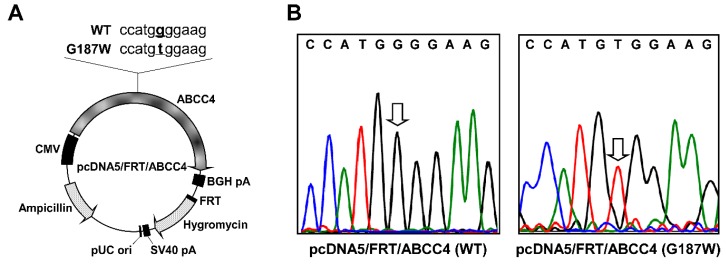
(**A**) Schematic illustration of the pcDNA5/FRT/ABCC4 vector. The partial cDNA sequences at position 554 to 564 in ABCC4 (WT) and (G187W) cDNA are indicated. The base at position 559 is indicated by bold text with underline. ABCC4, ABCC4 cDNA; BGH, bovine growth hormone; CMV, cytomegalovirus; pA, polyadenylation sequence; pUC ori, pUC vector origin of replication; SV40, simian virus 40. (**B**) Electropherograms confirming nucleotide substitution (G > T) at position 559 in ABCC4 (WT) cDNA. The nucleotide at position 559 is indicated by arrow.

**Figure 3 cells-08-00039-f003:**
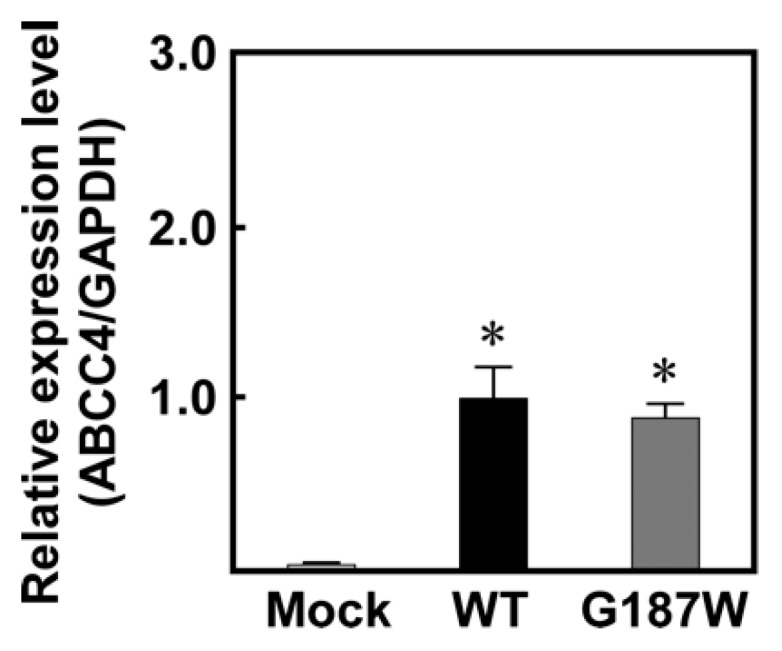
Levels of *ABCC4* mRNA in cells established using the Flp-In™ system. The levels of *ABCC4* and *GAPDH* mRNA were measured using qPCR with specific primer sets for *ABCC4* and *GAPDH*, as described in Materials and Methods. Data are calculated as ratios relative to the *GAPDH* mRNA levels in the cells and normalized to the ratio of *ABCC4*/*GAPDH*. Data are expressed as mean values ± S.D. (n = 5). Statistical analyses were performed using one-way ANOVA and Tukey’s HSD test (* *p* < 0.01 compared to the Mock group).

**Figure 4 cells-08-00039-f004:**
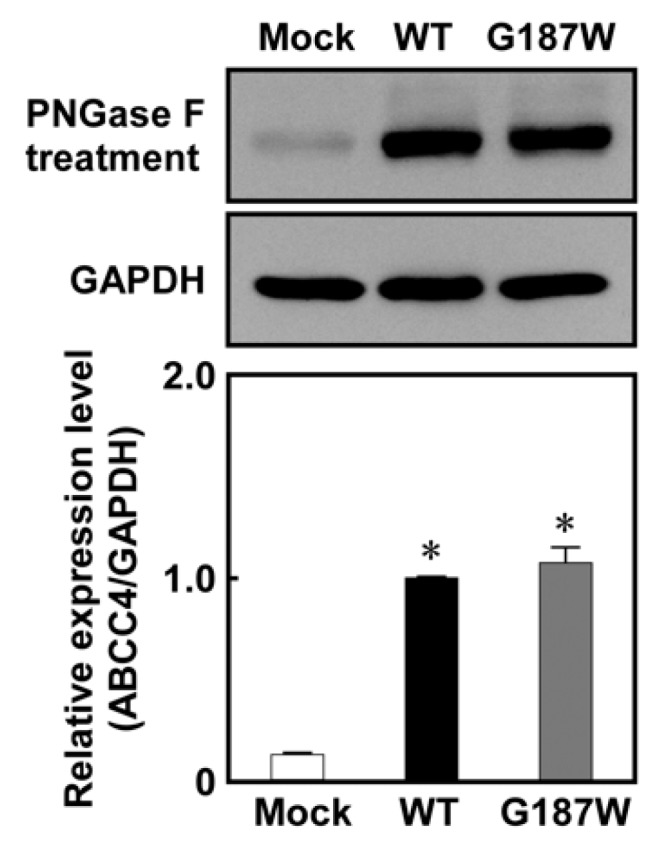
Levels of ABCC4 in cells established using the Flp-In™ system. ABCC4 and GAPDH levels were determined by western blotting analysis with specific antibodies for ABCC4 and GAPDH and their levels were quantified using ImageJ (Wayne Rasband, Bethesda, MD, USA) as described in Materials and Methods. ABCC4-specific monoclonal antibody (M4I-10) or GAPDH-specific antibody was used for protein detection in PNGase F-treated cell lysate. The experiments were performed independently more than two times. Data are expressed as mean values ± S.D. (n = 3). Statistical analyses were performed using one-way ANOVA and Tukey’s HSD test (* *p* < 0.01 compared to the Mock group).

**Figure 5 cells-08-00039-f005:**
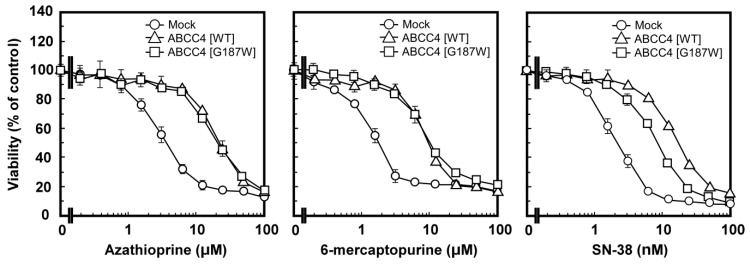
Anticancer drug resistance profiles of cells established using the Flp-In™ system. The anticancer drug resistance profiles of the cells were evaluated using the 3-(4,5-dimethyl-2-thiazol-2-yl)-2,5-diphenyl-2*H*-tetrazolium bromide (MTT) assay, as described in Materials and Methods. Data are expressed as mean values ± S.D. (n = 3–4).

**Table 1 cells-08-00039-t001:** Summary of allelic frequencies of major SNPs of human *ABCC4* in the Japanese population.

Rs No.	Reference Allele Frequency	Alternative Allele Frequency
**Human Genetic Variation**		
rs11568658 (G187W)	0.8755	0.1245
rs2274407 (K304N)	0.7506	0.2493
rs3765534 (E757K)	0.8465	0.1536
**Integrative Japanese Genome Variation Database**		
rs11568658 (G187W)	0.8621	0.1379
rs2274407 (K304N)	0.7384	0.2615
rs3765534 (E757K)	0.8457	0.1543

Data acquired from the Human Genetic Variation (Human Genetic Variation 2018) and Integrative Japanese Genome Variation Database (Integrative Japanese Genome Variation Database 2018).

**Table 2 cells-08-00039-t002:** Anticancer drug resistance profiles (EC_50_) of the cells.

Compounds	EC_50_		
	Mock Cells	ABCC4 (WT) Cells	ABCC4 (G187W) Cells
azathioprine	5.1 ± 0.4 μM	21.5 ± 1.0 * μM	22.2 ± 2.0 * μM
6-mercaptopurine	2.2 ± 0.07 μM	9.6 ± 1.3 * μM	10.6 ± 0.5 * μM
SN-38	2.1 ± 0.2 nM	16.3 ± 1.6 * nM	8.9 ± 1.1 *, ** nM

SN-38, 7-ethyl-10-hydroxy-camptothecin. The drug resistance profiles of cells established using the Flp-In™ system were evaluated, as described in Materials and Methods. Data are expressed as mean values ± S.D. (n = 3–4). Statistical analyses were performed using one-way ANOVA and Tukey’s HSD test: * *p* < 0.01 compared to the Mock group; ** *p* < 0.01 compared with the wild-type (WT).

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
