# Peer review of "A Human ABC Transporter ABCC4 Gene SNP (rs11568658, 559 G > T, G187W) Reduces ABCC4-Dependent Drug Resistance"

_cells, 2019, doi:10.3390/cells8010039_

Round 1

Reviewer 1 Report

This paper is identical to reference 10, a paper of the same scientific group. The results are publishable, but the manuscript presents a lot of parts directly copied from that reference 10.

I suggest later publishing considering the experiments you claim to perform at the discussion (lines 267-270), as you also say in ref 10.

Other considerations:

Line 45: In reference 5, they do not use azathioprine, but other thiopurines.

Define abbreviations: DMEM, FBS.

Lines 111 and 112: eliminate doble parenthesis.

In M&M, paragraphs 2.62.7 and 2.8 are identical to reference 10. So it is not necessary to describe those methods. use "as describe in ...".

Line 162: Last parenthesis deleted.

Line 175: As you have copied it from reference 10, the plural in Arrows and locations is wrong.

The discussion is almost identical that in ref 10. In line 245 you claim "These observations indicated that ABCC4 cDNA transfected into Flp-In-293 cells was functional and we could examine the effects of SNPs in the ABCC4 gene on ABCC4-mediated drug resistance in transfected cells". This observation was already done in ref 10.

Author Response

Response to Reviewer 1’s Comments:

We greatly appreciate the constructive comments of Reviewer 1 regarding our manuscript (cells-404869) entitled “A human ABC transporter ABCC4 gene SNP (rs11568658, 559 G>T, G187W) reduces ABCC4-dependent drug resistance.” written by Megumi Tsukamoto, et al.

We carefully considered the questions, instructions, and comments obtained from you and the two reviewers, and then performed review of the data and revised the manuscript accordingly.  Our responses to the comments of you are enclosed below.  Before submission, the revised manuscript has been checked by Editage (www.editage.jp) which offers professional English language editing service.

In the revised manuscript, the sentences and the references altered to respond to the reviewer’s comments and edited by Editage have been shown in red characters with and without comment, respectively.

Reviewer 1

Comments and Suggestions for Authors

1.      This paper is identical to reference 10, a paper of the same scientific group. The results are publishable, but the manuscript presents a lot of parts directly copied from that reference 10. I suggest later publishing considering the experiments you claim to perform at the discussion (lines 267-270), as you also say in ref 10.

Response:

We appreciate the reviewer’s comment with suggestion.  The data showing the levels of intracellular accumulation of test drugs must help us to understand the molecular mechanisms by which substitution of Gly187 to Trp in ABCC4 alters the drug resistance profiles of cancer cells.  On the other hand, the data showing the levels of ABCC4 on the plasma membrane is mandatory in order to fully understand the molecular mechanisms by which substitution of Gly187 to Trp in ABCC4 alters the drug resistance profiles of cancer cells.  The experiments for those data are currently in progress in our laboratory, where three major SNPs in Japanese population (G187W, K304N, and E757K) are investigated.  Although we can combine those data with the present results in future, we will not because the study will take at least a year to be completed.  The present results are enough to indicate the possibility that prognosis associated with SN-38 chemotherapy would be better in cancer patients positive for expression of ABCC4 (G187W) than ABCC4 (WT).  We believe that some other researchers who read the present paper will start any studies to investigate the clinical impact of ABCC4 (G187W) on prognosis associated with SN-38 chemotherapy.  As such opportunities come early, the present paper should be published soon.

2.      Line 45: In reference 5, they do not use azathioprine, but other thiopurines.

Response:

We appreciate the reviewer’s comment with instructions.  According to reviewer’s instructions, the reference 10 was cited in place of the reference 5 at line 43 in the revised manuscript.

3.      Define abbreviations: DMEM, FBS.

Response:

We appreciate the reviewer’s comment with instructions.  According to reviewer’s instructions, the abbreviation list was prepared and added to line 300 in the revised manuscript.

4.      Lines 111 and 112: eliminate doble parenthesis.

Response:

We appreciate the reviewer’s comment with instructions.  According to reviewer’s instructions, doble parenthesis were eliminated in the revised manuscript.

5.      In M&M, paragraphs 2.62.7 and 2.8 are identical to reference 10. So it is not necessary to describe those methods. use "as describe in ...".

Response:

We appreciate the reviewer’s comment with suggestion.  As Reviewer 1 pointed out, paragraphs 2.6, 2.7, and 2.8 are not necessary to be described because they are identical to those in reference 10.  On the other hand, the existence of those sentences will make the reader easy to understand the experiments performed in the present study.  To response to reviewer’s comment with suggestion, new sentences were added to paragraphs 2.6, 2.7, and 2.8 in the revised manuscript. 

6.      Line 162: Last parenthesis deleted.

Response:

We appreciate the reviewer’s comment with instructions.  The parenthesis Reviewer 1 pointed out should not be deleted because the description of ABCC4 variant has been unified such as "ABCC 4" by using parentheses in our previous and present studies.  Therefore, the last parenthesis in line 162 in the original manuscript was not deleted in the revised manuscript.

7.      Line 175: As you have copied it from reference 10, the plural in Arrows and locations is wrong.

Response:

We appreciate the reviewer’s comment with instructions.  According to reviewer’s instructions, those description was corrected in the revised manuscript.

8.      The discussion is almost identical that in ref 10. In line 245 you claim "These observations indicated that ABCC4 cDNA transfected into Flp-In-293 cells was functional and we could examine the effects of SNPs in the ABCC4 gene on ABCC4-mediated drug resistance in transfected cells". This observation was already done in ref 10.

Response:

We appreciate the reviewer’s comment.  To response to reviewer’s comment, the original sentences in lines 241-248 in the original manuscript were modified in lines 251-262 in the revised manuscript.

  We hope that the revised manuscript will satisfactorily meet the questions, assignments, and comments obtained from the reviewers, and be accepted for publication in the Cells.  Should you have any questions, please do not hesitate to contact me (hnakagaw@isc.chubu.ac.jp).

I am looking forward to hearing your final decision in your earliest convenience.

Sincerely yours.

Hiroshi Nakagawa, Ph.D.

Reviewer 2 Report

I suggest publication of The manuscript. Nevertheless last conclusion regarding patients should be eliminated since this study design Is not appropriate to state this statment 

Author Response

Response to Reviewer 2’s Comments:

We greatly appreciate the constructive comments of Reviewer 1 regarding our manuscript (cells-404869) entitled “A human ABC transporter ABCC4 gene SNP (rs11568658, 559 G>T, G187W) reduces ABCC4-dependent drug resistance.” written by Megumi Tsukamoto, et al.

We carefully considered the questions, instructions, and comments obtained from you and the two reviewers, and then performed review of the data and revised the manuscript accordingly.  Our responses to the comments of you are enclosed below.  Before submission, the revised manuscript has been checked by Editage (www.editage.jp) which offers professional English language editing service.

In the revised manuscript, the sentences and the references altered to respond to the reviewer’s comments and edited by Editage have been shown in red characters with and without comment, respectively.

Reviewer 2

Comments and Suggestions for Authors

1.      I suggest publication of The manuscript. Nevertheless last conclusion regarding patients should be eliminated since this study design Is not appropriate to state this statment.

Response:

We appreciate the reviewer’s comment with instructions.  According to reviewer’s instructions, the sentences in line 26-29 and line 272-274 in the original manuscript were modified in line 26-27 or removed in the revised manuscript, respectively.

  We hope that the revised manuscript will satisfactorily meet the questions, assignments, and comments obtained from the reviewers, and be accepted for publication in the Cells.  Should you have any questions, please do not hesitate to contact me (hnakagaw@isc.chubu.ac.jp).

I am looking forward to hearing your final decision in your earliest convenience.

Sincerely yours.

Hiroshi Nakagawa, Ph.D.

Reviewer 3 Report

In this study, the impact of ABCC4 rs1156858 (G187W, one of the major non-synonymous SNP variants in the Japanese population) on drug resistence profiles was evaluated. The authors have shown that the substitution of Gly at position 187 of ABCC4 to Trp resulted in reduced SN-38 resistence with potential association of this SNP with SN-38 efficiency. A subscribed manuscript is capable of being published depending on minor revision process. 

Major points:

Abstract: authors should mention the frequency of rs11568658 in the Japanese population in the abstract of this manuscript.

Methods, paragraph 2.2: Authors should include Figure of the confirmed nucleic acid sequence of ABCC4 (G187W) variant as they estimated by direct sequencing with gene analyzeers. This sequence confirms the successful preparation of ABCC4 variant plasmid.

Minor points:

Methods: paragraph 2.4 How was the quality of total RNA samples verified?

Methods: paragraph 2.5 Authors should mention type of mRNA quantification in methodological part of this manuscript.  

Author Response

Response to Reviewer 3’s Comments:

We greatly appreciate the constructive comments of Reviewer 1 regarding our manuscript (cells-404869) entitled “A human ABC transporter ABCC4 gene SNP (rs11568658, 559 G>T, G187W) reduces ABCC4-dependent drug resistance.” written by Megumi Tsukamoto, et al.

We carefully considered the questions, instructions, and comments obtained from you and the two reviewers, and then performed review of the data and revised the manuscript accordingly.  Our responses to the comments of you are enclosed below.  Before submission, the revised manuscript has been checked by Editage (www.editage.jp) which offers professional English language editing service.

In the revised manuscript, the sentences and the references altered to respond to the reviewer’s comments and edited by Editage have been shown in red characters with and without comment, respectively.

Reviewer 3

Comments and Suggestions for Authors

In this study, the impact of ABCC4 rs1156858 (G187W, one of the major non-synonymous SNP variants in the Japanese population) on drug resistence profiles was evaluated. The authors have shown that the substitution of Gly at position 187 of ABCC4 to Trp resulted in reduced SN-38 resistence with potential association of this SNP with SN-38 efficiency. A subscribed manuscript is capable of being published depending on minor revision process.

Major comments

1.      Abstract: authors should mention the frequency of rs11568658 in the Japanese population in the abstract of this manuscript.

Response:

We appreciate the reviewer’s comment with suggestion.  According to reviewer’s suggestion, the frequency of rs11568658 in the Japanese population was added to the abstract in the revised manuscript.

2.      Methods, paragraph 2.2: Authors should include Figure of the confirmed nucleic acid sequence of ABCC4 (G187W) variant as they estimated by direct sequencing with gene analyzeers. This sequence confirms the successful preparation of ABCC4 variant plasmid.

Response:

We appreciate the reviewer’s comment with suggestion.  According to reviewer’s suggestion, a new figure and a legend were added to the revised manuscript as Figure 2.  As a result, figure number was changed in the revised manuscript.

Minor comments

1.      Methods: paragraph 2.4 How was the quality of total RNA samples verified?

Response:

We appreciate the reviewer’s question.  In the present study, the quality of total RNA was verified by measuring absorbance at 260 and 280 nm and calculating the absorbance ratio (A260/A280).  The A260/A280 values of total RNA prepared in the present study were comparable.  First strand cDNA was prepared from total RNA by using random hexamers as a primer.  The quality of first strand cDNA was verified by comparing threshold cycles among samples analyzed in Quantitative real-time PCR.  In order to answer the reviewer’s question, original sentences at lines 104-107 and 168-169 were modified at line 102-107 and 171-175 in the revised manuscript, respectively.

2.      Methods: paragraph 2.5 Authors should mention type of mRNA quantification in methodological part of this manuscript.

Response:

We appreciate the reviewer’s instruction.  According to reviewer’s instruction, original sentence at lines 104-107 and 168-169 were modified at line 102-107 and 171-175 in the revised manuscript, respectively.

We hope that the revised manuscript will satisfactorily meet the questions, assignments, and comments obtained from the reviewers, and be accepted for publication in the Cells.  Should you have any questions, please do not hesitate to contact me (hnakagaw@isc.chubu.ac.jp).

I am looking forward to hearing your final decision in your earliest convenience.

Sincerely yours.

Hiroshi Nakagawa, Ph.D.

Round 2

Reviewer 1 Report

This paper is OK for publishing at the present form.